# Determination of Bio-Based Fertilizer Composition Using Combined NIR and MIR Spectroscopy: A Model Averaging Approach

**DOI:** 10.3390/s22155919

**Published:** 2022-08-08

**Authors:** Khan Wali, Haris Ahmad Khan, Mark Farrell, Eldert J. Van Henten, Erik Meers

**Affiliations:** 1Farm Technology Group, Wageningen University & Research, 6708 PB Wageningen, The Netherlands; 2CSIRO Agriculture and Food, Kaurna Country, Locked Bag 2, Glen Osmond, SA 5064, Australia; 3Department of Green Chemistry and Technology, University of Gent, 9820 Merelbeke, Belgium

**Keywords:** Near-infrared (NIR) and Mid-infrared (MIR) spectroscopy, bio-based fertilizers, partial least square regression, wavelength selection, model averaging

## Abstract

Application of bio-based fertilizers is considered a practical solution to enhance soil fertility and maintain soil quality. However, the composition of bio-based fertilizers needs to be quantified before their application to the soil. Non-destructive techniques such as near-infrared (NIR) and mid-infrared (MIR) are generally used to quantify the composition of bio-based fertilizers in a speedy and cost-effective manner. However, the prediction performances of these techniques need to be quantified before deployment. With this motive, this study investigates the potential of these techniques to characterize a diverse set of bio-based fertilizers for 25 different properties including nutrients, minerals, heavy metals, pH, and EC. A partial least square model with wavelength selection is employed to estimate each property of interest. Then a model averaging, approach is tested to examine if combining model outcomes of NIR with MIR could improve the prediction performances of these sensors. In total, 17 of the 25 elements could be predicted to have a good performance status using individual spectral methods. Combining model outcomes of NIR with MIR resulted in an improvement, increasing the number of properties that could be predicted from 17 to 21. Most notably the improvement in prediction performance was observed for Cd, Cr, Zn, Al, Ca, Fe, S, Cu, Ec, and Na. It was concluded that the combined use of NIR and MIR spectral methods can be used to monitor the composition of a diverse set of bio-based fertilizers.

## 1. Introduction

With an ever-increasing world population, the demand for food is growing at an alarming rate [1]. To meet this increasing demand for food, agricultural production thrived on the use of chemical fertilizers during the past few decades to improve soil fertility and help increase crop yield [2]. However, the use of chemical fertilizers is not without drawbacks. Chemical fertilizers are based on limited natural resources (e.g., phosphorus P) or energy-consuming chemical processes (e.g., Nitrogen N). The excessive use of chemical fertilizers can also negatively impact the environment through eutrophication and acidification [3,4].

To address the negative impacts of synthetic fertilizer, modern agriculture is exploring the application of bio-based fertilizers. Bio-based fertilizers have a long history of use in agriculture, including compost, manure, and bio-solids. However, during the post world war 2 (WW2) era, an increasing focus is on chemical fertilizers due to their effectiveness in raising crop production by rapid release of nutrients, abundant and low-cost availability, and ease of application [5]. Most bio-based fertilizers, on the other hand, have slow-release characteristics that may reduce the risk of environmentally deleterious losses. Bio-based fertilizers contain not only major nutrients (nitrogen (N), phosphorous (P), and potassium (K)) but also contain macro-nutrients magnesium (Mg), calcium (Ca) and iron (Fe). Bio-based fertilizers have the potential to improve crop yield and improve soil quality by maintaining organic matter [6,7].

The application of bio-based fertilizers for plant growth and soil fertility enhancement has many positive attributes, while excessive application may lead to pollution. Depending on the origin, bio-based fertilizers might also contain elements arsenic (As), boron (B), cadmium (Cd), chromium (Cr), copper (Cu), fluorine (F), lead (Pb), manganese (Mn), mercury (Hg), molybdenum (Mo), nickel (Ni), selenium (Se) and zinc (Zn) [8,9,10]. If added in excess, these elements are toxic both for human health and the environment [10]. One major drawback of bio-based fertilizers over their synthetic counterparts is that the nutrients are not readily available for plant uptake. Without risking the over-application of one or more key elements, information on their composition ( concentration of nutrients and their plant-available forms) is needed. This information about composition will allow agronomists to further assess whether crop-specific, region-specific, and season-specific fertilizers are feasible [11].

The nutrient concentration of bio-based fertilizers can be quantified by traditional chemical methods of analysis, which often involve digestion, titration, and distillation processes. Although these chemical analysis methods are considered accurate and precise, the steps involved in these chemical analysis methods are time-consuming and involve the use of different chemicals and complex laboratory equipment [12]. To acquire the composition information of bio-based fertilizers quickly, accurately, and economically, nondestructive and pollution-free techniques are needed.

Non-destructive testing (NDT) refers to the assessment process of materials for physical/chemical analysis without altering the original attributes of the sample. Typically, such methods require less sample preparation time and are therefore a cost-effective means of testing if accuracy can be maintained [13,14]. Spectroscopy techniques like near-infrared (NIR) and mid-infrared (MIR) are an economic alternative when multiple analyses and samples are required because they do not require expensive and time-consuming sample preprocessing, provided that the appropriate calibrations exist [15]. For all of these reasons, spectroscopic techniques have the potential for real-time characterization of bio-based fertilizers at the time of application to soil.

NIR spectroscopy has been investigated previously for characterization in agro-industrial compost [16], food [17,18], soils analysis [19,20,21], plants sciences [22], pharmaceuticals [23]. The application of NIR spectroscopy to bio-based materials (manure) has been reported by Ye et al. [24] for the prediction of N, P, K, and NH_3_-N content. Huang et al. [25] reported the feasibility of the prediction of (K, Ca, Mg, Fe, and Zn) in animal manure compost products using NIR. Recently, Bedin et al. [26] reported the use of NIR to predict the C, N, P, and K content in poultry litter with improved prediction accuracy, while Baldock et al. [27] found that NIR was capable of predicting the decomposition parameters of a wide range of bio-based fertilizers. NIR spectroscopy is considered a flexible technique due to the availability of chemometric evaluation tools and software, and the often limited sample preparation required [28]. Although NIR spectroscopy is a powerful tool and can be used to determine many properties within a wide range of analyses, it does have limitations. An example is the poor detection of phosphates and other inorganic P compounds due to the weak dipole moment of P–O [29]. Similarly, Heavy metals content is spectrally inactive in the NIR region, but can be predicted indirectly by correlating them with Fe, Al and Mn oxides and organic matter [30,31]. Wu et al. [30] concluded that the indirect correlation is only possible at higher concentration (≥1000 mg kg^−1^).

To potentially overcome some of these problems associated with NIR spectroscopy, Mid-infrared (MIR) spectroscopy which ranges from 2500–4000 nm, can be used for the characterization of different materials where NIR fails to produce acceptable results. MIR spectroscopy has been shown to give a better estimation of some soil properties compared with NIR spectroscopy [32,33]. Reeves et al. [34] reported an effective prediction of many soil properties using MIR spectroscopy. However, MIR has limitations for reliable analysis in non-homogeneous samples, and the size of the sample greatly affects its prediction capability [35]. Rourke et al. [36] and Javadi et al. [37] concluded that NIR and MIR spectroscopy techniques individually have the potential to predict certain elements with improved prediction accuracy while not predicting others. Hence, a potentially improved multi-sensor fusion approach could be realized by taking advantage of both NIR and MIR spectroscopy to quantify a range of nutrients in bio-based fertilizers.

To use the useful information present in two or more spectral sensors, different approaches like spectral fusion and model averaging can be utilized. Spectral fusion means combining spectra from two or more sensors and is expected to improve prediction accuracy. Previously, Wang et al. [31] showed that spectral fusion through concatenation of portable X-ray fluorescence (pXRF) with visible near-infrared (Vis-NIR) can improve the prediction capability for total carbon (TC) and total nitrogen (TN). A similar approach has been proposed by Aldabaa et al. [38] and Chakraborty et al. [39] for soil analysis, where the fused spectral model outperformed the models that utilized individual spectral information.

The fusion of spectral information from NIR and MIR can increase the useful information about a particular component, as well as amplify redundant and unwanted signals. The inclusion of unwanted signals therefore sometimes results in a reduction in prediction capability and can make the prediction model more complex [36,40]. The redundant and unwanted signals can be eliminated by using different wavelength selection techniques. In wavelength selection techniques only those wavelengths in the NIR and MIR spectral range, which are highly correlated with the response variables are selected and the unwanted signals are discarded. This selection of useful wavelengths is expected to improve the individual prediction from both sensors as well as reduce the redundancy in information.

Alternatively, the model averaging technique proposed by Granger et al. [41] can be used to overcome the problem associated with spectral fusion [36,42]. In the model averaging the individual prediction results from different sensors is combined and is expected to enhance prediction due to the complementary nature of the sensors used.

For use of bio-based fertilizers, a non-destructive measurement technique is needed that can determine a wide range of components, like nutrients, their plant-available forms as well as some minerals and heavy metals with sufficient accuracy and precision. A literature review revealed: (1) that NIR and MIR-based techniques could be used to that end, but will not be sufficient on an individual basis. Improved performance is expected from the fusion of information from both sensors. (2) so far NIR and MIR-based techniques have been investigated on a very limited set of chemical constituents of a limited set of different bio-based fertilizers.

Thus, this work contributes by sensing the contents of 25 nutrients in different bio-based fertilizers (manure, bio-solids, plant residues, and composts) using NIR and MIR spectroscopy. Furthermore, for improvement of estimations, a wavelength selection method followed by the model averaging technique is investigated to get the benefits of fusing the results from NIR and MIR sensors.

## 2. Materials and Methods

A dataset of 85 amendments was taken from Farrel et al. [43], described in Baldock et al. [27]. The data set describes 85 different bio-based fertilizers, including 50 composts from different composting facilities across Australia, 6 manure samples from different animals (cow, pig), 10 fresh plant residues derived from the major Australian crop species and some alternative species, and 19 biosolids obtained from a range of urban and rural wastewater plants. These spectral datasets enabled the testing of the robustness of prediction models using NIR and MIR spectra and model averaging.

### 2.1. Chemical Analysis

Data on the nutrient content and other chemical properties of the 85 bio-based fertilizers were obtained using standardized chemical analysis [43]. Briefly, pH and electrical conductivity (EC) were quantified using standard electrodes in a solid: water slurry (1:5 *w*/*v*). Total nitrogen (N) was quantified by high-temperature combustion analysis (Leco CNS 2000, Leco Corporation, St Joseph, MI, USA). The available free amino acid N (FAA-N) and ammonium-N (NH4-N) were quantified in 1:5 *w*/*v* water extracts by fluorimetry and colorimetry on a multimode plate reader (Synergy MX, Biotek, Winooski, VT, USA) using the methods of Jones et al. [44], Miranda et al. [45], and Mulvaney et al. [46], respectively. Total major and minor elements were quantified by inductively coupled plasma-mass spectroscopy (ICP-MS; 7500cx, Agilent Technologies, CA, USA) following HClO_4_/HNO_3_ digestion in open digestion tubes in a heated block [43]. The summary of the different properties of the 85 bio-based fertilizer samples is shown in Figure 1. The summary shows that heavy metals and trace elements concentration is for elements Ni, Zi, Mn, Se, Pb, Mo. Cr, Cu, Cd, and As is very low (less than 1000 mg kg^−1^) in the selected samples. The lower concentration of these elements might make it difficult to measure through NIR and MIR as suggested by Wu et al. [30].

### 2.2. Sample Characterization

The 25 properties of interest presented in this paper belong to three broad categories. (1) Essential plant nutrients (N, P, K) and plant-available forms of nitrogen (Nitrate (NO_3_-N)free amino acid (FAA-N) and NH_4_-N), (2) other elements and micronutrients (S, Al, As, Ca, Cu, Na, Fe, Mg, Mo, Ni, Mn, Zn, Cr, Co, Cd, Pb, and Se) and (3) properties pH and EC. The total data set (*n* = 85) was divided into a training set (80% i.e., *n* = 67) and test set (20% i.e., *n* = 18) according to Table 1 to ensure homogeneity over the train and test sets.

From the chemical distribution shown in Figure 1, it can be seen that the nutrient content between the sample groups has an obvious gradient distribution, which can well represent bio-based fertilizers in practical applications. The distribution of the concentrations of the elements is skewed, and the high variation of the concentration of elements in the data set enabled us to analyze why the predictions of some samples are better than others and how the concentration of an element changes the prediction capabilities of the model.

## 3. Model Development

Partial least squares regression (PLS) is one of the most widely used multivariate prediction methods in chemometric analysis. PLS projects spectral data into latent variables that explain the variances within the spectral data. Given a spectral matrix X and the corresponding truth matrix Y, PLS is used to find the scores (*T* and *U*) with loading (*P* and *Q*) and error matrix (*F*) from the decomposition of *X* and *Y* as given in Equations (Equation 1) and (Equation 2):(1)$$X=TP^{\prime }+F_{x}Y=UQ^{\prime }+F_{y}$$

While the original space relation is:(2)Y=XB+E
where matrix *B* is the regression coefficient and *E* is the residuals matrix. After the selection of characteristics wavelengths, the partial least squares regression model was established. The model was created using the optimal number of latent variables (lvs). The calibration set data was used to find the optimal number of latent variables, and the model obtained was used to predict the prediction set data.

### 3.1. Data Prepossessing

Pre-processing of NIR and MIR spectra is considered an important part of any quantitative or qualitative analysis [47,48]. Performing spectroscopy in the laboratory or in the field is often influenced by noise. This noise can reduce the signal-to-noise ratio (SNR) of the spectral information and, therefore, negatively affect a calibration model’s accuracy. Other challenges associated with NIR/MIR spectra include complex backgrounds and baselines, which introduce unwanted variations in the spectra and make calibration of the model complicated [49]. To deal with these problems the spectra are often pre-processed before any analysis. In the present study, the data set was mean-centered and pre-processed for baseline offset followed by a second-order polynomial de-trending algorithm. No further preprocessing was performed as this might have a negative impact on the prediction performance [50,51].

### 3.2. Optimal Wavelength Selection

In NIR and MIR spectroscopy, it is a challenge to identify upfront or prior to the wavelength bands, which will contain most of the information about the response variable [52]. Therefore all wavelengths are measured in the full NIR and MIR range. Sensing the whole range of wavelengths automatically also leads to the inclusion of irrelevant or less informative wavelengths. This inclusion of irrelevant or less informative signals has a negative impact on the prediction ability of the model and also might make the model unnecessarily complex [42,53,54]. In addition, the inclusion of this irrelevant information makes model interpretation difficult. Therefore, it is challenging to determine in advance which wavelengths or combinations are responsible for estimating the property of interest [52]. Hence measurement of the full NIR and MIR range, and then the identification and selection of a combination of wavelengths that contain information about response variables (nutrient content) are expected to improve prediction performance [55]. The wavelength selection can also help interpret fingerprint regions across the NIR and MIR spectral data which correspond to each response variable.

A simple method proposed by Frenich et al. [56] based on the PLS regression coefficients (B) is used in this paper for the selection of characteristic wavelengths. The method proposes that the value of B can be used as a measure of importance for an individual wavelength in the prediction of the response variable. This is similar to the interpretation of parameters in linear regression. A high absolute value of *B* indicates that the corresponding wavelength $\lambda_{i}$ is more important and has a high correlation with the response variable and vice versa [42].

The method of wavelength selection using B is implemented in three steps. First, the PLS model is fitted and optimized in the entire spectrum to find the optimum latent variables (lvs). The latent variables are optimized by observing the mean square error (MSE) as shown in Figure 2. The optimum number of lvs are the ones where the MSE of cross-validation is minimum.

PLS regression coefficients are extracted and their absolute values are arranged in ascending order. The NIR spectra and the corresponding regression coefficients for nitrogen as a response variable are shown in Figure 3 and Figure 4.

In the second step, the wavelengths (for each spectra) are sorted using the indices as corresponding to the sorted absolute value of the PLS regression coefficient.

In the third step, wavelengths that had a low B value and low correlation with the response variable are discarded using Algorithm 1. The algorithm iterate and discard one wavelength at a time (the one with the lowest absolute value of the associated regression coefficient) and rebuild the calibration model and evaluate the mean square error (MSE) of the cross-validation set. At some point, removing wavelengths will increase the MSE, and that is the stopping criterion for the optimization algorithm. The remaining wavelengths were selected and it was expected that they will improve the prediction performance of the model. The wavelength selected for the nitrogen content in NIR spectra is shown in Figure 5.
**Algorithm 1** Wavelength selection algorithm  1: Fit PLS model on NIR/MIR spectral data  2: Find MSE of cross-validation (CV)  3: Store MSE as MSE(0) for the start of the loop  4: Find all the regression coefficients(B)  5: Arrange B in ascending order  6: Arrange spectra accordingly  7: **procedure**
wavelength selection(λ)  8:     initialize i = 1  9:     Discard one wavelength at time λ(i)10:     Fit PLS on remaining wavelengths11:     Find MSE of CV12:     **if** MSE(i)≤MSE(i−1) **then**13:         Discard Wavelength14:         i = i + 115:         Repeat step 916:     **else**17:         Stop18:     Print all the discarded wavelengths19:     Print all the remaining wavelengths20:     Selected wavelengths = remaining wavelengths

Note: The remaining number of wavelengths must be greater than or equal to optimize the number of lvs.

### 3.3. Model Averaging

In the model averaging method, the results of the NIR and MIR spectral analysis are combined to improve the prediction results as proposed by Granger et al. [41] and shown in Figure 6. The proposed method uses ordinary least squares regression to utilize covariance structure in the prediction errors, where the weighting attributed to the prediction result of each sensor does not necessarily sum to one [36]. The wavelength selection algorithm is applied individually on NIR and MIR to select the characteristics wavelengths and then the prediction results from each sensor are combined using Equation (Equation 3). To the predictions obtained from each individual sensor, weights are assigned according to their performances in the training set. The results from NIR spectra get a higher weight if it has a lower RMSE value compared to MIR spectra and vice versa as shown in Equation (Equation 3).
(3)$$Y_{i}^{\prime }=W_{0}+\left(W_{1}\cdot Y_{NIR}^{\prime }\right)+\left(W_{2}\cdot Y_{MIR}^{\prime }\right)$$
where $Y_{i}^{\prime }$ contains the observed vector of the response variable (element of interest), $W_{o}$ is the intercept, $Y_{NIR}^{\prime }$ and $Y_{MIR}^{\prime }$ are the individual prediction results of the NIR and MIR spectral models, and $W_{1}$ and $W_{2}$ are the weights assigned to the NIR and MIR predictions, respectively. Ordinary least squares (OLS) regression is used to find the values of $W_{o}$, $W_{1}$, and $W_{2}$. For model development, the prediction results of training and test data sets from NIR and MIR were concatenated, resulting in a two-column feature matrix.

### 3.4. Model Assessment Criteria

For model assessment, the performance parameters, root mean square error (RMSE), correlation coefficient R^2^, and the ratio of performance deviation (RPD) were used. R^2^ shows the goodness of fit between the predicted value and the experimental value. As proposed by Saeys et al. [57], a value for R^2^ (0.66∼0.80) indicates approximate quantitative predictions, whereas a value for R^2^ (0.81∼0.90) reveals good prediction. Calibration models having R^2^ > 0.90 are considered to be excellent. RPD is defined as the standard deviation of the predicted value divided by the RMSE, which is a measure of the effectiveness and overall predictability of the regression model. According to Saeys et al. [57] and Zornoza et al. [58] RPD < 2 is considered insufficient for applications, whereas a value for RPD between 2 and 2.5 makes approximate quantitative predictions possible. For values between 2.5 and 3 predictions can be classified as good, and an RPD > 3 indicates an excellent prediction. RMSE is used to measure the deviation between the predicted value and the experimental value. The smaller the value of RMSE indicates a smaller deviation between the predicted value and the experimental value. The calculation of these parameters are as follows:(4)$$\mathrm{RMSE}=\sqrt{\frac{1}{n}\Sigma_{i=1}^{n}{\left(\frac{Y_{i}^{\prime }-Y_{i}}{\sigma_{i}}\right)}^{2}}$$
(5)$$R^{2}=1-\frac{\Sigma_{i=1}^{n}(Y_{i}-Y_{i}^{\prime }{)}^{2}}{\Sigma_{i=1}^{n}{\left(y_{i}-\bar{y}\right)}^{2}}$$
(6)$$\mathrm{RPD}=\frac{\mathrm{STD}(Y_{i})}{\mathrm{RMSE}}$$

Here $y_{i}^{\prime }$ and $y_{i}$ are the predicted and actual values of the response variables, $\bar{y}$ is the mean value of the actual value of the response variable, and $\mathrm{STD}(Y_{i})$ is the standard deviation of the actual response variables.

## 4. Results

### 4.1. Near-Infrared (NIR) and Mid-Infrared (MIR) Predictions

The prediction results before and after characteristic wavelength selection for each sensor (NIR and MIR) are presented in Table 2. The prediction results based on the full spectrum from both NIR and MIR for N, NH_4_-N, Al, P and EC were better, However, for the metal and mineral contents, the results were not satisfactory in the current study. It can be observed that the results of wavelength selection outperformed the results based on the full spectrum for all elements. The essential plant nutrients (N, P, and plant-available forms of nitrogen) are predicted relatively better than the rest of the elements. Nitrogen has the highest (R^2^ = 0.94) followed by aluminum (R^2^ = 0.92), phosphorous, and ammonium ion, while K was predicted more poorly. The prediction results of N, FAA-N, NO_3_, pH, Cr, Cu, Se, Ca, Mn, and P were better in the NIR range, while the predictions of NH_4_-N, Ec, As, Cd, Zn, Al, Fe, K, Mg, Na, and S was better in the MIR range in both cases (with and without wavelength selection). For Co, Mo, Ni, and Pb, the results of the prediction of NIR and MIR were comparable, though MIR results were slightly better than NIR. The ranking in Table 2 was established for each sensor by observing the RMSE, R^2^, and RPD values for each sensor. The sensor having the lowest RMSE, and highest R^2^ and RPD values are preferred for the prediction of a particular nutrient. The table shows the ability of individual sensors and the ranking of each sensor in predicting the nutrient contents. For Na, Zn, Ni, Mo, Cr, Co, Cd, As and Mn, the prediction of NIR and MIR did not reach an acceptable range, i.e., (R^2^ < 0.7) even with wavelength selection. This is due to the fact that these elements are featureless in NIR and MIR range [30]. They are mostly indirectly predicted using NIR and MIR spectroscopy. In terms of predictive performance with R^2^ > 0.7, 8 elements were predicted to an acceptable range using NIR spectral method, while 9 elements reached to an acceptable range using MIR spectral method. The results in Table 2 suggests that MIR performed better for Al, and Fe. and that is why the prediction performances of metal content is better in MIR range [36].

If the result with R^2^ > 0.7 is an acceptable prediction for a particular response variable, then a total of 17 out of 25 elements were sufficiently predicted with wavelength selection, as shown in Table 2. Improvement in prediction results is expected by combining the results from both NIR and MIR using model averaging as proposed by Rourke et al. [36] and are presented in the next section.

### 4.2. Prediction of Elements Using Model Averaging NIR and MIR Results

The combined results from NIR and MIR prediction using model averaging are shown in Table 3. The wavelength selection algorithm is applied individually to spectral data of NIR and MIR and then the prediction results are combined using Equation (Equation 3). The percent improvement in prediction from both NIR and MIR sensors indicates that model averaging is a good technique for combining the prediction results. The percent improvement in Table 3 shows that model averaging improved the prediction of Zn, Al, Cr, Cd, Ca, and Fe substantially in terms of RMSE, R^2^, and RPD from both NIR and MIR individual results. A positive improvement in prediction results was observed for all properties compared to the results obtained from individual sensor predictions.

For elements Pb, K, Cu, Cr, Mn, As, Cd, and Co, 0.75 ≤ R^2^ < 0.81 was observed using model averaging. The prediction result for elements Cr, Co, Cd, As and Mn using model averaging reached to an acceptable range (R^2^ > 0.7). Major and trace elements (Ni, Zn, Mo, and Na) were difficult to predict using individual senor results and model averaging couldn’t improve their prediction to acceptable range. The unreliable predictions of Ni, Zn, Mo, and Na present a barrier for the combine use of NIR and MIR for the quantification of composition of bio-based fertilizers. If the result with R^2^ > 0.7 is considered an acceptable prediction as proposed by Saeys et al. [57], then a total of 21 out of 25 elements are predicted with wavelength selection and model average, as shown in Table 3. overall, the reasonable to good prediction of most nutrients, trace elements and metal contents in the current study using model averaging of NIR and MIR results suggests that measurement of full suite composition of bio-based fertilizers might be possible if other sensors are combined with NIR and MIR.

## 5. Discussion

The potential of NIR and MIR spectroscopy was investigated both in full range, as well as selected wavelengths, the range for estimation of bio-based fertilizers composition. The results based on the full range of NIR and MIR spectrum were encouraging for some essential nutrients (N, NH4, Al, and P) but could not produce promising results for other elements. The prediction results for the full range of NIR and MIR spectrum, suggest that a total of 13 properties were predicted to an acceptable range (R^2^ > 0.70) [57]. The poor results might be due to the irrelevant information included in the spectral range [42,53,59] which makes the calibration model complex. Therefore, the wavelength selection technique for each response variable resulted in improved prediction from both NIR and MIR full range. The improvement in the prediction performance can be viewed in terms of RMSE, R^2^, and RPD as shown in Table 1. Prediction results from selected wavelength enabled the prediction of 17 elements out of 25 to an acceptable range in the current study. The prediction results and the corresponding ranking in Table 2 suggest that NIR can produce better results for certain elements while MIR can be useful for others. By combining both NIR and MIR using model averaging outperformed both the individual results as shown in Table 3.

The model averaging of results obtained from individual sensors improved the prediction for each response variable. Maximum improvement in terms of RMSE, R^2^ and RPD are observed for Cd, Co, Cr, and Mn which were not predicted to an acceptable range according to the criteria of R^2^ > 0.7 by individual sensors. For Al, Ca, Fe, Mg, S, NH_4_-N substantial improvements were observed. The prediction results for both individual sensors and the model averaging for Ni, Zn, Mo, and Na did not reach an acceptable range (R^2^ > 0.70), although, substantially improved from individual sensor predictions [57]. The lower prediction of metal content was expected as they are spectral inactive in NIR and MIR range. Their predictions are only possible by linking them with other properties which show more features in NIR and MIR range [59]. As proposed by Wang et al. [30] NI, Zn, Mo, and other metals contents can indirectly be predicted only if their concentration is not less than 1000 mg kg^−1^. Thus the lower concentration of these elements in the current study might be another reason for their poor predictions.

The set of sample in the current study is diverse and contain four different sources, the correlation between metal content and the spectrally active element might be different in each source. This can affect the indirect prediction of some of the metal elements and result in their poor prediction. In order to overcome the problem associated with NIR and MIR alternate sensors can be investigated in future studies [36]. Alternative sensors namely, X-ray fluorescence (XRF) and Fourier transform infrared photoacoustic spectroscopy (FTIR-PAS) sensors might be more effective in predicting these properties [60,61].

Overall, model averaging improved the prediction of all the elements of interest in the current study. The results shown in Table 3, demonstrate that 21 out of 25 properties were predicted using the model averaging strategy. This improvement for the detection of nutrients and other elements can be compared with the results obtained for NIR and MIR in the literature. Huang et al. [25] evaluated different nutrients and elements (N, Fe, Mg, Ca) in manure using NIR and, in comparison, the model averaging perform better in terms of R^2^ in the current study, despite the fact that the samples selected in the current study contain four variants (manure, bio-solids, plant residues, and composts).

## 6. Conclusions

In this study, a wide range of nutrients, their plant-available forms, minerals, and heavy metal contents are quantified using NIR and MIR spectroscopy in a diverse set of bio-based fertilizers. A wavelength selection technique is applied for the selection of characteristic wavelengths, and then the Individual prediction capabilities of NIR and MIR are investigated for quantification of nutrient contents. A model averaging technique that combines model outcomes derived from NIR and MIR was then used which resulted in an improved prediction performance predicting 21 out of 25 nutrients and other properties. The most notable improvement in prediction was obtained for Cd, Cr, Zn, Al, Ca, Fe, S, Cu, Ec, and Na. However, for Ni, Zn, Mo, and Na, the obtained prediction results from model averaging could not reach the acceptable range (R^2^ > 0.70); although it improved substantially from individual sensors predictions. Therefore, combining the results from NIR and MIR spectral methods using model averaging is well placed to replace traditional wet chemical analysis methods for the analysis of bio-based fertilizers composition. In the future, we plan to investigate the potential of NIR and MIR with other sensors (XRF, and FTIR-PAS) to provide more comprehensive coverage of bio-based fertilizer composition.

## Figures and Tables

**Figure 1 sensors-22-05919-f001:**
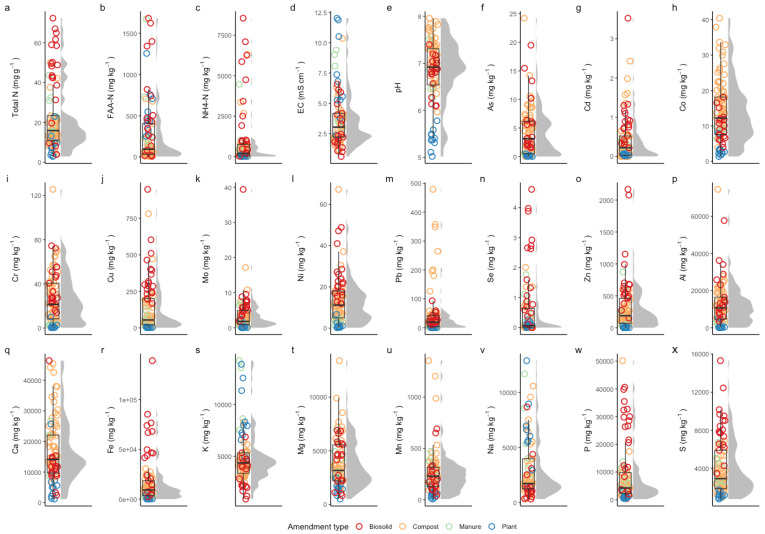
Chemical distribution of plants’ essential nutrients (N, P, K and plant-available form of nitrogen), and total elements derived through chemical analysis from bio-solid (red-color circles), composts (orange-color circles), manure (green-color circles) and plants residues (blue-color circles). y-axis shows the concentration of each element with all units in mg kg^−1^ except N which is in mg g^−1^ and the number of each element is shown on the x-axis. (**a**–**x**) Different properties of the 85 bio-based fertilizer samples.

**Figure 2 sensors-22-05919-f002:**
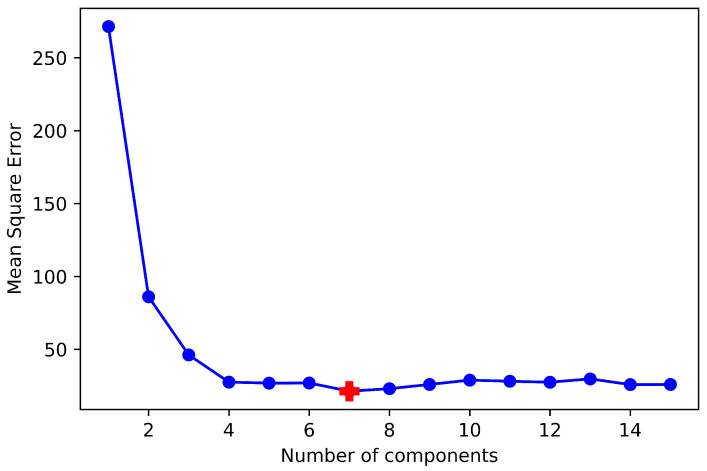
Mean square error as a function of the number of latent variables (without variable selection for nitrogen content).

**Figure 3 sensors-22-05919-f003:**
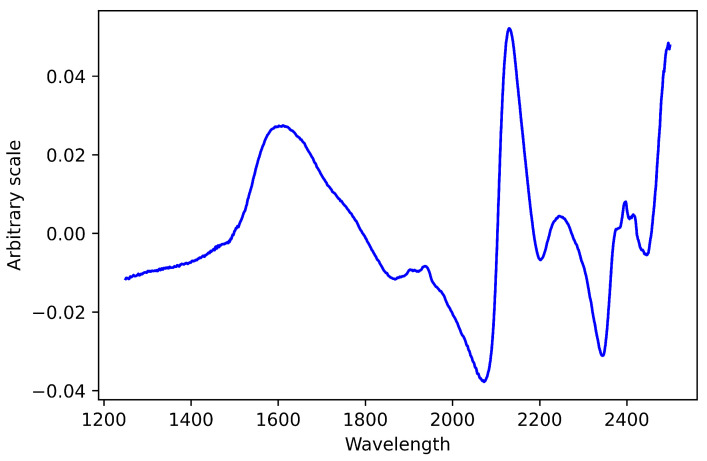
NIR pretreated spectrum.

**Figure 4 sensors-22-05919-f004:**
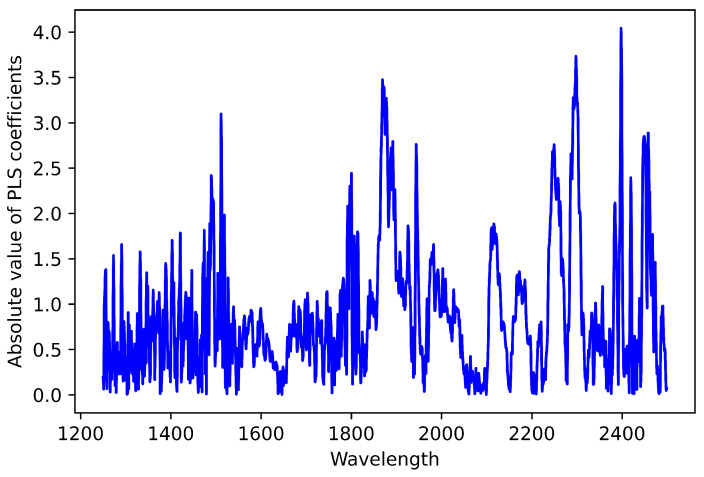
PLS regression coefficients absolute values against each wavelength for nitrogen contents.

**Figure 5 sensors-22-05919-f005:**
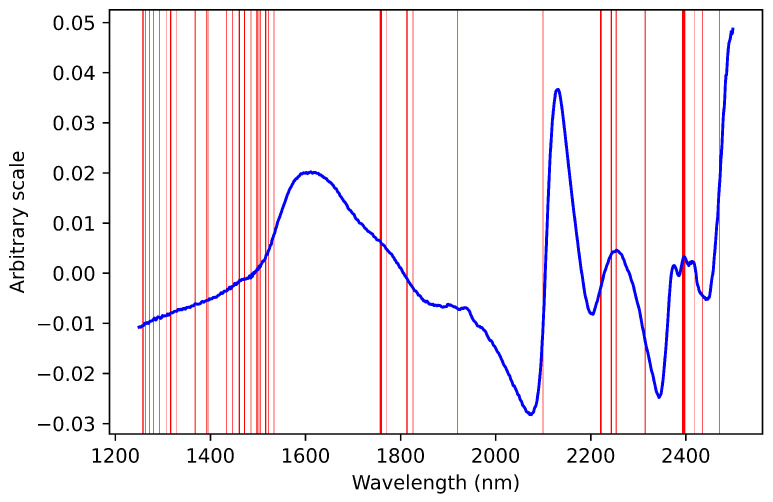
Selected bands of wavelengths for nitrogen contents.

**Figure 6 sensors-22-05919-f006:**
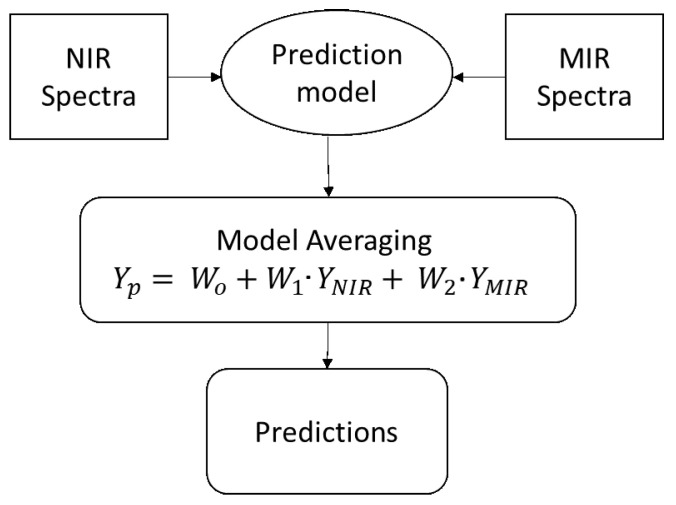
Model averaging for NIR and MIR prediction.

**Table 1 sensors-22-05919-t001:** Train test split from each sample.

Type	Total	Train	Test
composts.	50	40	10
manure.	6	4	2
plants residues.	10	8	2
bio-solids.	19	15	4

**Table 2 sensors-22-05919-t002:** The goodness of fit for essential nutrients for plants (N. P, K and plant-available form of nitrogen) and total elements derived from near-infrared (NIR) and mid-infrared (MIR) with and without wavelength selection. All units are g kg^−1^, except Cd, As, Cr, Se, Mo, and Ni in mg kg^−1^.

	Prediction Results without Wavelength Selection						Prediction Results with Wavelength Selection						
	**NIR**			**MIR**			**NIR**			**MIR**			
**Element**	**RMSE**	**R^2^**	**RPD**	**RMSE**	**R^2^**	**RPD**	**RMSE**	**R^2^**	**RPD**	**RMSE**	**R^2^**	**RPD**	**Ranking**
N	3.92	0.88	2.86	4.87	0.85	2.3	2.88	0.94	6.94	3.56	0.92	5.4	NIR
FAA-N	0.46	0.68	1.96	0.54	0.63	1.81	0.24	0.79	2.21	0.27	0.78	2.12	NIR
NO_3_-N	0.60	0.66	1.74	0.62	0.61	1.58	0.4	0.73	2.48	0.59	0.70	2.18	NIR
NH_4_-N	0.71	0.82	2.43	0.63	0.87	2.85	0.66	0.88	2.87	0.59	0.92	3.69	MIR
EC	1.23	0.8	2.31	1.12	0.82	2.56	1.13	0.85	2.61	1.08	0.86	2.72	MIR
pH	0.31	0.78	2.21	0.43	0.71	1.92	0.27	0.82	2.38	0.35	0.76	1.83	NIR
As	3.38	0.63	1.8	3.35	0.67	1.86	3.06	0.68	1.66	3	0.7	1.82	MIR
Cd	0.45	0.47	1.32	0.39	0.54	1.51	0.37	0.54	1.47	0.32	0.63	1.66	MIR
Co	7.43	0.61	1.7	7.54	0.63	1.73	5.59	0.66	1.78	5.6	0.67	1.79	MIR/NIR
Cr	17.17	0.53	1.37	21.01	0.47	1.31	13.76	0.67	1.69	18.11	0.56	1.5	NIR
Cu	0.13	0.63	1.43	0.15	0.57	1.37	0.09	0.72	1.89	0.097	0.7	1.81	NIR
Mo	10.21	0.09	0.97	10.06	0.11	0.98	8.81	0.12	1.08	8.76	0.15	1.1	MIR/NIR
Ni	10.13	0.39	0.92	10.07	0.41	0.98	8.7	0.45	1.12	8.6	0.47	1.2	MIR/NIR
Pb	0.09	0.67	1.91	0.081	0.71	1.98	0.043	0.75	2.03	0.042	0.76	2.06	MIR/NIR
Se	0.51	0.79	2.64	0.54	0.75	2.61	0.38	0.87	2.82	0.39	0.86	2.72	NIR
Zn	0.36	0.26	1.11	0.25	0.34	1.23	0.27	0.32	1.5	0.18	0.41	1.55	MIR
Al	7.8	0.75	2.11	6.12	0.83	3.24	5.5	0.84	2.5	3.8	0.92	3.68	MIR
Ca	6.13	0.73	2.09	6.38	0.67	1.96	4.3	0.82	2.39	4.72	0.78	2.18	NIR
Fe	15.56	0.61	1.74	12.23	0.76	2.48	12.88	0.73	1.98	9.91	0.84	2.58	MIR
K	1.82	0.65	1.87	1.78	0.71	1.93	1.67	0.76	2.03	1.62	0.77	2.1	MIR
Mg	1.83	0.67	1.89	1.79	0.71	1.94	1.56	0.75	2.03	1.53	0.76	2.05	MIR
Mn	0.31	0.61	1.78	0.33	0.58	1.67	0.17	0.73	1.98	0.18	0.66	1.74	NIR
Na	2.12	0.43	1.31	2.21	0.53	1.39	2.014	0.56	1.51	1.95	0.59	1.56	MIR
P	3.54	0.86	3.23	3.92	0.81	2.89	2.67	0.93	3.8	3.12	0.9	3.24	NIR
S	1.38	0.76	2.33	1.36	0.78	2.45	1.08	0.83	2.46	1.05	0.85	2.57	MIR

**Table 3 sensors-22-05919-t003:** The goodness of fit for major nutrients (N. P, K and plant-available form of nitrogen), and other properties derived from model averaging of NIR and MIR prediction results. All units are g kg^−1^, except Cd, As, Cr, Se, Mo, and Ni in mg kg^−1^.

	Model Averaging			Percent Improvement from NIR			Percent Improvement from MIR		
**Element**	**RMSE**	**R^2^**	**RPD**	**RMSE**	**R^2^**	**RPD**	**RMSE**	**R^2^**	**RPD**
N	2.84	0.96	6.98	−1.39	2.13	0.58	−20.22	4.35	29.26
FAA-N	0.22	0.81	2.34	−8.33	2.53	5.88	−18.52	3.85	10.38
NO3-N	0.37	0.76	2.57	−7.50	4.10	3.60	−37.28	8.57	17.88
NH4-N	0.57	0.94	3.8	−13.64	6.82	32.40	−3.39	2.17	2.98
EC	0.99	0.89	2.97	−12.39	4.71	13.79	−8.33	3.49	9.19
pH	0.25	0.85	2.6	−7.41	3.66	9.24	−28.57	11.84	42.08
As	2.91	0.75	2.1	−4.90	10.29	26.51	−3.00	7.14	15.38
Cd	0.27	0.75	2.02	−27.03	38.89	37.41	−15.63	19.05	21.69
Co	4.84	0.75	1.98	−13.42	13.64	11.24	−13.57	11.94	10.61
Cr	9.82	0.77	2.08	−28.63	14.93	23.08	−45.78	37.50	38.67
Cu	0.079	0.8	2.21	−12.22	11.11	16.93	−18.56	14.29	22.10
Mo	8.34	0.16	1.2	−5.33	33.33	11.11	−4.79	6.67	9.09
Ni	8.58	0.48	1.21	−1.38	6.67	8.04	−0.23	2.13	0.83
Pb	0.038	0.81	2.32	−11.63	8.00	14.29	−9.52	6.58	12.62
Se	0.35	0.89	3.06	−7.89	2.30	8.51	−10.26	3.49	12.50
Zn	0.14	0.53	1.67	−48.15	65.63	11.33	−22.22	29.27	7.74
Al	3.18	0.94	4.12	−42.18	11.90	64.80	−16.32	2.17	11.96
Ca	3.22	0.9	3.19	−25.12	9.76	33.47	−31.78	15.38	46.33
Fe	9.56	0.85	2.68	−25.78	16.44	35.35	−3.53	1.19	3.88
K	1.54	0.8	2.2	−7.78	5.26	8.37	−4.94	3.90	4.76
Mg	1.36	0.81	2.28	−12.82	8.00	12.32	−11.11	6.58	11.22
Mn	0.16	0.75	2.02	−5.88	2.74	2.02	−11.11	13.64	16.09
Na	1.77	0.66	1.71	−12.12	17.86	13.25	−9.23	11.86	9.62
P	2.51	0.94	4.31	−5.99	1.08	13.42	−19.55	4.44	33.02
S	0.81	0.91	3.33	−25.00	9.64	35.37	−22.86	7.06	29.57

## Data Availability

Not applicable.

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
