# Peer review of "Determination of Bio-Based Fertilizer Composition Using Combined NIR and MIR Spectroscopy: A Model Averaging Approach"

_sensors, 2022, doi:10.3390/s22155919_

Round 1
Reviewer 1 Report
the present work aims to sense the content of 25 nutrients in different biobased fertilizers (manure, bio-solids, plant residues, and composts). A wavelength selection method followed by the model averaging technique is investigated to get the benefits of fusing NIR and MIR sensor results.
They investigated the potential of these techniques to chara
terize a diverse set of biobased fertilizers for 25 properties including nutrients, minerals, heavy metals, pH, and EC. A partial least square model with wavelength selection was employed to estimate each property of interest.
Most notably the improvement in 14 prediction performance was observed for Cd, Cr, Zn, Al, Ca, Fe, S, Cu, Ec, and Na.
The paper is well written with a highly scientific language with reasonable arrangement. the modeling part is sounded and presented briefly with dissection.
In my opinion, the work is highly recommended for publication in the present section Sensing and Imaging of Sensors Journal
Only some minor comments
-In text and Figure 2 please define the mean square error (MSE) and add it to the caption
-Please define the correct Y-axis in Figures 3 and 5
-What do you mean by absorbance spectra!!!?
-All figure resolution should be improved
Author Response
We are extremely thankful for the reviewer's suggestions, concerns, and questions about our paper. The comments were really helpful in shaping the paper. The comments, concerns of the reviewers, and the answer to those comments are attached in the Reviewer 1 comments file.

Reviewer 2 Report
The ms sensors-1798605 with the title of Determination of Bio-based Fertilizer Composition Using Combined NIR and MIR Spectroscopy: A Model Averaging Approach is investigating the interesting topic, but the ms has to be significantly improved before it can go for further steps. See my comments below:
Keywords are missing! The authors have to be very careful when writing scientific research.
Please cite the uncited text particularly the introduction section.
L24-26 please cite these relevant and recent ref: https://doi.org/10.1016/j.resconrec.2019.104647 /
L40-44 usually the biofertilizers do not have such all heavy metals elements, particularly pb, as. Organic fertilizers can have, but bio-fertilizers do not. Therefore, please revise these sentences.
The introduction should be reduced in the text as well as the authors should give citations for uncited text.
In material and methods: The authors should descript the study and materials in details instead of summarizing them.
Some figures are low quality, please improve
L275-276 Why the prediction of NIR and MIR did not reached to the acceptable range of Na, Zn, Ni, Mo, Cr, Co, Cd, As and Mn? Because they were low? Or something else?
The authors should avoid using common words or nonscientific words such as etc. or goodness or others.
The discussion section is very poor and the authors should significantly improve it and make it deeply written.
Best regards, Reviewer
Author Response
We are extremely thankful for the reviewer's suggestions, concerns, and questions about our paper. The comments were really helpful in shaping the paper. The comments, concerns of the reviewers, and the answer to those comments are provided in the attached file.

Reviewer 3 Report
The authors use the NIR and MIR spectroscopy to determine the bio-based fertilizer composition with a model averaging approach, and got a good result, which is useful for future application for not only bio-based fertilizer but also other areas. However, one issue has to be addressed before it is published.
The NIR spectrum consists of overtones and combination vibrations of molecules that contain CH, NH, or OH groups. In this work, a lot of metal elements of Cd, Zn, Al, Fe, K, Mg, Na, ……, were determined. Because these metals cannot bond with hydrogen, how the quantitative information of these metal elements was extracted from the NIR spectra.
Other minor questions
1. page 2, line 1, “post-WW2 increasing……”, give the whole words “WW2).
2. page 7, Figure 3 is not the NIR raw spectrum, it should be the pretreated spectrum.
3. Figure 4, The title of the Y-axis is incorrect.
4. Figure 5, Change the “Absorbance spectra” to “Arbitrary scale”.
5. L240, “2” in “R2” should be superscript.
6. Table 3, Change the last NIR to MIR.
Author Response
We are extremely thankful for the reviewer's suggestions, concerns, and questions about our paper. The comments were really helpful in shaping the paper. All the concerns and questions are addressed in the attached file.

Round 2
Reviewer 2 Report
The ms has been improved.